# How does company misconduct affect company survival risk?—evidence from China

Ling Lu [1,2]*

**1** Postdoctoral Workstation, Jiangxi University of Finance and Economics, Nanchang, China, **2** School of Management, Jiujiang University, Nanchang, China

* lulu_995179@163.com

## Abstract

Based on the data of China's listed companies from 2000 to 2022, this study investigates how company misconduct affect company survival risk through survival analysis. The research results show that company misconduct significantly increases company survival risk. After considering endogeneity issues and conducting robustness tests, this conclusion still holds. Mechanism analysis reveals that company misconduct significantly increases enterprise survival risk by reducing investor confidence and increasing corporate financing constraints. Further analysis considering company and regional heterogeneity, which shows that non-state-owned, small-scale, low equity concentration, and eastern region company face more severe survival risk after misconduct. The findings extend the research on the influencing factor of company survival, and provide new empirical evidence for revealing how corporate misconduct affect company survival risk.

**Citation:** Lu L (2024) How does company misconduct affect company survival risk?—evidence from China. PLoS ONE 19(12): e0306767. https://doi.org/10.1371/journal.pone.0306767

**Data Availability Statement:** All relevant data are within the paper and its Supporting Information files.

**Funding:** This paper is supported by National Natural Science Foundation of China (72263021)

## Introduction

In 2019, the sudden outbreak of the COVID-19 pandemic swept across the globe, causing unprecedented disruptions on the economies worldwide. Particularly for developing countries, the impact of the pandemic is more severe due to their fragile economic foundations, poor healthcare conditions, and insufficient emergency response capabilities. As a vital component of the economy, the development status of enterprises directly affects the prosperity or decline of national economies. The spread of the pandemic leads to disruptions in the supply chains of enterprises, which hinders the supply of raw materials and components, and results in production halts. Simultaneously, with the reinforcement of pandemic prevention and control measures, profound changes occurs in people's consumption habits and patterns. As a consequence, the revenue of enterprises sharply declines, and plunges many enterprises into cash flow constraints and substantial financial pressures. Facing with such daunting scenario, numerous enterprises are compelled to halt operations or reduce production capacity, which leads to a decrease in national employment rates, and further exacerbates economic stagnation and social instability. Hence, mitigating enterprise survival risks, improving their operational environments, and thereby safeguarding the health of national economies and fostering social

found; China Postdoctoral Science Foundation (2020M682104); University Humanities and Social Science Research Projects of Jiangxi Province (JJ22210). The funders had no role in study design, data collection and analysis, decision to publish, or preparation of the manuscript.

**Competing interests:** The authors have declared that no competing interests exist.

stability are pressing issues that governments and the business community worldwide urgently need to address.

Previous studies concentrate on the connotation and measurement of enterprise survival [1–5], and the key determinants of enterprise survival. The key determinants focus on enterprise internal factors and external environmental factors. The internal factors mainly contain enterprise features such as size, age, ownership structure and ESG performance [6–9], enterprise activities such as export and innovation [10–13]. The external environmental factors mainly contain geographical location, macroeconomic fluctuation, financial market structures, market competition, industrial policies, government subsidies, etc. [14–18]. However, despite considerable attention paid to influencing factors of corporate survival risks, insufficient focus has been directed to company misconduct. The precise pathways through which company misconduct affects the risk of corporate survival remain unclear and thus empirical examination is needed.

This paper takes data from A-share listed companies in China between 2000 to 2022, and adopts survival analysis method to explore how company misconduct affect company survival risk. First, non-parametric KM estimation and Clog-log discrete time survival mode are employed to analyze the extent to which violation affect the probability of failure of listed companies. Second, this paper uses mediation effect model to investigate the impact mechanism through which company misconduct affect survival risk. Third, this paper considers company and regional heterogeneity, and finds that non-state-owned, small-scale, low equity concentration, and eastern region companies face more severe survival risk after misconduct.

In comparison to previous literature, this paper may have following contributions. First, it studies the influencing factor of company survival and heterogeneity, which enriches the research of relative topics. There is little attention to whether company misconduct leads to company survival risk. This study aims to reveal the survival risks of company misconduct, and exams the heterogeneity of the impact of company misconduct on survival from the perspectives of both the firm and the region features, and provides evidences for different companies which face more survival risk after misconduct. Second, this paper researches how company misconduct affect company survival risk, that is explores the path through which corporate misconduct affects survival, and enriches the research perspectives on transmission mechanisms of company misconduct. Third, most studies on company survival in China mainly use the China Industrial Enterprise Database, with the research subjects being all state-owned and large scale non-state-owned industrial enterprises. Relevant research on listed companies is very limited. This paper uses the data from Chinese listed companies, and explores listed companies survival risk after going public through survival analysis method.

The subsequent structure of this paper is arranged as follows: the second part is literature review and research hypotheses; the third part is data and method; the fourth part is empirical results analysis; the fifth part is endogeneity treatment and robustness tests; the sixth part is heterogeneity analysis; the seventh part is discussion; the eighth part is summary and recommendations.

## Literature review and research hypotheses

### The connotation and measurement of company survival

The survival time of a company is commonly defined by scholars as the period from its inception to its exit from the market. If a company remains operational throughout the observation period, it is deemed to have sustained survival; however, if it exits from the market during the observation period, it is considered a failure [1, 2, 19, 20]. Some scholars link company survival risk with financial distress, positing that when companies experience situations such as debt

default, bankruptcy liquidation, insufficient liquidity, or cash flow interruption, they confront survival risks [21, 22]. Some researchers use various indicators including financial accounting index, market information index, macroeconomic indicators, and corporate governance mechanisms index to construct or predict financial distress [23–27]. The most representative one of all is Altman's Z-score bankruptcy prediction model, frequently employed to analyze the risks of business failure and financial distress of companies from different sectors and economic regimes [3–5, 28–31]. Additionally, some scholars employ two years of consecutive declines in market value, three years of consecutive losses in profits and company delisting as measures of company survival failure [32–34].

## Key determinants of company survival

Numerous scholars analyse influencing factors of company survival from various perspectives, and they can be categorized into internal and external factors. Concerning internal factors, scholars study the influence of size and age of company survival, and the empirical literature shows that large and old age firms are better positioned for survival due to enhanced resource access, stable clientele, and economies of scale [35, 36]. However, this relationship depends on the stage of the industry's life cycle [37, 38]. Furthermore, scholars investigate the impact of innovation on corporate survival. Drawing from industrial organization theory, innovation is posited as intrinsic to corporate survival, with innovative firms establishing and maintaining competitive advantages in the market, thereby significantly extending their longevity [12–14, 39–41]. However, viewpoints suggest a complex relationship between innovation and survival, and this relationship is moderated by various factors such as internal company features and industry disparities, which leads to considerable uncertainty regarding the relationship between innovation and survival duration [42–44]. Regarding export behavior, scholars argue that export provides opportunities for firms to acquire advanced production technology and management expertise, thereby fostering sustained operations [10, 45, 46]. Nonetheless, as newcomers to export markets, exporting firms may face discrimination from foreign governments, consumers, and enterprises, which place them at a disadvantage in the competitive export market and consequently increasing survival risk [11, 47, 48]. Additionally, scholars also study the influence of equity structure, financing constraints, dual circulation, and ESG performance on corporate survival [6–8, 49–53].

External environmental factors primarily encompass macroeconomic elements, market competition, and geographical location. Concerning macroeconomic factors, scholars primary concern about economic fluctuations, fiscal policies, financial frictions, and labor quality as significant determinants influencing corporate survival. These factors can directly affect the company both from market inputs and outputs, thereby affecting its risks in the market and survival capabilities [15, 16, 54–57]. Regarding market competition, while some studies posit that market competition reduces the risk of corporate survival [13, 58], others find that competition either escalates survival risk or has no significant impact on survival risk, and the relationship is also contingent upon factors such as company features, industry features, and governance mechanisms, which modulates its effects [59, 60]. Geographical location and industrial agglomeration are also pivotal factors which affect corporate survival. Companies, which are situated in different geographical locations and surrounded by varying industrial clusters, face divergent levels of competition and resource accessibility, which leads to either positive or negative externalities that impact their survival risk [17, 18, 61–64].

## The consequences of company misconduct

Company misconduct damages the reputation of the company, which is detrimental to company's competition and long-term development. Meanwhile, company engaged in misconduct may also face significant financial costs such as fines and litigation fees. By studying the consequences of corporate misconduct, many scholars believe that being investigated and punished for misconduct will have a significant impact on the company's value, market reactions, commercial credit, loan costs, investment levels, risk-taking behavior, and executive changes [65–69]. Some scholars study the negative reputation effects of violation penalties, and find that penalties imposed on corporations can lead to a certain degree of negative impact on their reputations, and it exacerbates their financing difficulties and agency conflicts [70, 71]. The tarnished image of corporations further affects their reputation ratings and business collaborations, and that results in decreased profitability [72], thereby putting them at a disadvantage in market competition. However, some scholars study the effect of misconduct on restaurant firm survival, and find that the survival probability of a restaurant firm increases as its own misconduct increases, but spreading peer misconduct which triggered by own misconduct will deteriorate the business environment and threaten the survival of the firm [73].

## Gaps and research hypotheses

Few studies focus on the impact of company misconduct on company survival risk [73]. Moreover, scant attention has been paid to whether such effects exhibit heterogeneity, and there is a lack of theoretical exploration and empirical testing concerning the mechanisms through which misconduct influences company survival. Additionally, existing research focuses on the restaurant firm, which leads to the lack of researching on publicly listed companies. As the cornerstone of Chinese capital market and the primary contributors to Chinese economic growth, publicly listed companies' survival risk after misconduct need to be discussed.

Previous studies find that company misconduct has adverse effects on company from the aspects of reputation, commercial credit, financing cost, etc., and exacerbates the financing dilemma and agency conflict of enterprises. Thus, the following research hypothesis is proposed.

H0. Misconduct by listed company increases company survival risk.

When a company is punished or regulated due to misconduct, it may face public criticism, investigations, fines and rectifications. In severe cases, the managers may even be criminally detained, and it directly affects the normal operation of the company. At the same time, negative news is transmitted externally, and causes damage to the company's reputations. Consequently, it diminishes the capital market's expectations of the enterprise, which causes investors to lose confidence and no longer trust the products offered by the company. Feroz et al. [74] investigates 58 companies with disclosure inaccuracies in U.S. and finds that these companies' stock prices fall by 13% within two days after the disclosure of their misconduct. Nourayi [75] finds that the severity of corporate misconduct is inversely related to stock returns. Chen and Gao [76] analyse the characteristics of company misconduct and their market reactions in Chinese securities market from 1999 to 2001, and find that the securities market exhibits negative reactions to misconduct announcements, which leads to negative abnormal returns for company stocks. This result has been approved by Chen et al., Feng and Xu [77]. Jiang and Zhao [78] explore misconduct by company managers and find that the disclosure of managers' violation leads to a decline in company stock prices. The above researches indicate that negative news of corporate misconduct can undermine investor confidence in the capital market, thereby exposing companies at survival risk. Based on the above analysis, the following research hypothesis is proposed.

H1. Misconduct by listed company reduces investor confidence, thereby increasing company survival risk.

Negative impacts on corporate reputation will affect the financing activities of the company in the future, resulting in more severe financing constraints. When a company's misconduct is exposed, investors receive this negative signal and adjust their risk assessments of the company, doubting the company's operational risks and the authenticity of its financial statements. In this situation, it becomes difficult for the company to obtain the necessary funds, or the company has to pay higher financing costs in the capital market, resulting in financing constraints. Chen and Wang [79] study the relationship between corporate misconduct and commercial credit, and find that after being punished for illegal activities, companies would reduce commercial credit lines and significantly increase the cost of commercial credit. Liu and Dai [80] study loan financing under the background of corporate misconduct and found that compared to non-violating companies, violating companies have fewer loans, higher costs, and the increase in frequency and severity of violation leads to stricter financing constraints. Zhang et al. [81] find that corporate bond issuers with misconduct records need to pay higher issuance fees and coupon rates. Zhu [82] study the impact mechanism of misconduct on corporate financing and find that misconduct can damage companies' reputation, inducing them to turn to higher-cost lease financing and short-term loan financing. Some scholars study the influence of financial market development and financing constraints to the probability of corporate survival. It is generally believing that the less developed the financial market and the stronger the financing constraints, the lower the survival rate of companies [83, 84]. Based on the above analysis, the following research hypothesis is proposed.

H2. Misconduct by listed company increases financing constraints, thereby increasing company survival risk.

## Data and method

### Methodology and model specification

This study uses survival analysis, which differs from conventional empirical regression analysis. Survival analysis models can consider both the occurrence and non-occurrence of endpoint events, and survival time, and can deal with right-censored data and conduct multifactor analysis of firm survival. Scholars define the time from listing to delisting as the survival time of listed companies [32, 85]. However, China's capital market has always adhered to the principle of "strict entry and difficult exit", and the qualifications for listing are relatively scarce. Once a company loses its ability to operate continuously, the company and local government usually take various measures to avoid delisting. As a result, the delisting rate in China's capital market is very low. Some companies that lost their ability to operate continuously still remain in the capital market for a long time. From 1990 to 2023, there are only 177 companies delisted in China. Given the unique institutional background in China, the sample of delisted firms is limited. Following the approach of Song et al., Hu and Sheng [86, 87], this paper considers the first time being special treatment (including the first time being labeled ST or *ST) as the endpoint event in survival analysis, namely the death event. Initially, let T represent the duration from company listing to the first special treatment, and let the company survival function $S(t)$ represents the probability of maintaining a certain state beyond time t

(t = 1,2,3, . . .):

$$S(t) = p(T \geqq t) \tag{1}$$

Subsequently, Kaplan-Meier (KM) non-parametric estimation is used for descriptive analysis. The KM estimation is as follows:

$$\hat{S}(t) = \prod\nolimits_{t_i \leq t} \frac{n_i - d_i}{n_i} \tag{2}$$

$n_t$ is the number of individuals experiencing an risk event at time $t_i$ and $d_i$ is the number of individuals at risk at time $t_i$. By plotting the survival and hazard functions for firms with and without misconduct, an initial assessment of the impact of violation on firm survival can be made.

Next, the Clog-log survival risk model is used for a more systematic and rigorous estimation. The Cox proportional hazards semi-parametric estimation model which is based on continuous time and the Clog-log model which is based on discrete time are widely used in multifactor survival analysis. The Cox model is more flexible and can obtain reliable estimates of regression coefficients without making any restrictive assumptions about the baseline hazard function. However, this model requires that variables do not vary over time and assumes that event times are absolutely continuous, implying that events can occur at any point in time. But some risk variables that affect the survival of listed companies do change over time, and annual company data divides the company's survival time into discrete intervals. Therefore, it is more reasonable to consider a model for discrete time processes, that is Clog-log model. The model formulation in this study is as follow:

$$\begin{aligned} F(h_{it}) = Cloglog(1 - h_{it}) \quad &= log[-log(1 - h_{it})] \\ &= \beta_0 + \beta_1 \, Violate_{it} + \gamma Controls_{it} + \lambda_t + \eta_i + \varepsilon_{it} \end{aligned} \tag{3}$$

Subscript i and t represent individual companies and years, respectively. $h_{it} = 1 - exp[-exp(\gamma' X_{it} + \varphi_t)]$ represents the discrete time hazard rate of companies, with higher value of $h_{it}$ indicating higher survival risk for the firm. $\varphi_t$ represents the baseline hazard function, which is only dependent on time and does not vary with individual firms. $X_{it}$ includes independent variable and control variables. The estimated coefficient of the independent variable $\beta_1$ indicates the impact of violation on firm survival risk, which is the core coefficient in this study. This coefficient does not directly reflect the influence of variable and needs to be transformed into hazard ratio $e^{\beta_1}$. If the hazard ratio $e^{\beta_1} > 1$ (that is $\beta_1 > 0$), it indicates that the variable increases firm survival risk, and vice versa. $\lambda_t$ and $\eta_i$ represent the time and industry fixed effects, respectively, and $\varepsilon_{it}$ is the random error term.

## Research variables

The core independent variable in this study is whether the company misconduct (*Violate*). According to the penalty announcements from China Securities Regulatory Commission, Shanghai Stock Exchange and Shenzhen Stock Exchange, the main misconduct of Chinese listed companies include profit fabrication, asset overstatement, false recording, significant omissions, asset misappropriation, internal transactions, illegal buying, selling of stock, etc., totaling 15 situations. Following the method of Wei et al., Luo and Li [88, 89], whether a company is in violation is determined by whether there is a penalty announcement from regulatory authorities for that year. If there is a misconduct, the value of "*Violate*"is 1; if there is no violation, the value is 0.

Drawing on existing research, control variables in this study are mainly considered from three aspects: (1) financial and governance indicators of listed companies, including company size (*Asset*), company age (*Age*), liquidity ratio (*FA*), total asset turnover ratio (*Assetturn*), return on total assets (*Return*), financial leverage (*FL*), free cash flow (*FC*), asset growth rate (*Assetgrow*), Tobin's Q (*Q*), price-to-book ratio (*PB*), debt-to-asset ratio (*Dbtass*), ownership concentration (*Owncon*), and company board size (*Director*); (2) macroeconomic indicators: gross domestic product (*GDP*) of the province where the company is located; (3) other control variables, including dummy variable for company ownership(*Act*) with a value of 1 for state-owned company and 0 for others, dummy variable for company location (*ProvNum*) with a value of 1 for the eastern region and 0 for the central and western regions.

In summary, the variable name, description, data sources and descriptive statistics of research variables are shown in Tables 1 and 2.

## Data collection and processing

This paper takes data from A-share listed companies in China between 2000 to 2022, the end-point event is defined as the first time a company receives special treatment (including the first instance of being labeled ST or *ST). Since it takes at least one to two years after company goes public to observe the endpoint event, companies listed in 2022 and beyond are excluded. Due to the unique features of the financial industry, samples of listed companies in the financial industry are also excluded. Data of company misconduct is sourced from China Stock Market & Accounting Research Database (CSMAR) of company violation penalty announcements.

**Table 1. Variable name and description.**

| Variable Type | Variable Name | Variable Symbol | Variable Description | Data Sources |
|---|---|---|---|---|
| Independent variable | Company Misconduct | Violate | Takes a value of 1 if the company engages in violations, otherwise 0 | CSMAR |
| Control Variables | Company Size | Asset | Natural logarithm of total assets | CSMAR |
| | Company Age | Age | Years of establishment of company, calculated as (2022 - year of firm establishment) | RESSET |
| | Liquidity Ratio | FA | Current assets / current liabilities | CSMAR |
| | Total Asset Turnover Ratio | Assetturn | Total revenue / total assets | CSMAR |
| | Return on Total Assets | Return | Net profit / total assets | CSMAR |
| | Financial Leverage | FL | (Net profit + income tax expense + financial expenses) / (Net profit + income tax expense) | RESSET |
| | Free Cash Flow | FC | (Profit before interest and taxes + depreciation and amortization - increase in operating working capital - capital expenditure) / total assets | CSMAR |
| | Asset Growth Rate | Assetgrow | (Total assets at year-end - total assets at year-beginning) / total assets at year-beginning | CSMAR |
| | Tobin's Q | Q | (Market value of equity + net debt) / tangible asset value | CSMAR |
| | Price-to-Book Ratio | PB | Price per share / book value per share | CSMAR |
| | Debt-to-Asset Ratio | Dbtass | Total liabilities / total assets | CSMAR |
| | Ownership Concentration | Owncon | Percentage of shares held by the largest shareholder | CSMAR |
| | Company Board Size | Director | Number of members on the company board | CSMAR |
| | GDP of the Province Where the Company is Located | GDP | GDP of the province where the firm is located | CSMAR |
| | Industry | Ind | Industry dummy variable: According to the CSRC industry classification standard, excluding the financial industry, there are total 18 industries | CSMAR |
| | Company Ownership | Act | Ownership dummy variable: 1 for state-owned, 0 otherwise | RESSET |
| | Company Location | ProvNum | Location dummy variable: 1 for eastern region, 0 for central and western regions | RESSET |

**Table 2. Descriptive statistics of variables.**

| Variables | Observations | Mean | Standard Deviation | Minimum | Maximum |
|---|---|---|---|---|---|
| Violate | 41,564 | 0.707 | 0.455 | 0.000 | 1.000 |
| Asset | 41,564 | 22.042 | 1.296 | 18.897 | 25.931 |
| Age | 41,564 | 23.624 | 5.450 | 6.000 | 64.000 |
| FA | 41,564 | 2.529 | 2.583 | 0.260 | 15.423 |
| Assetturn | 41,564 | 0.637 | 0.432 | 0.060 | 2.592 |
| Return | 41,564 | 0.069 | 0.049 | -0.047 | 0.346 |
| FL | 41,564 | 1.397 | 0.989 | 0.555 | 7.181 |
| FC | 41,564 | -0.020 | 0.142 | -0.628 | 0.287 |
| Assetgrow | 41,564 | 0.224 | 0.381 | -0.384 | 2.271 |
| Q | 41,564 | 1.948 | 1.237 | 0.869 | 8.520 |
| PB | 41,564 | 3.582 | 3.106 | 0.609 | 22.817 |
| Dbtass | 41,564 | 0.418 | 0.202 | 0.053 | 0.990 |
| Owncon | 41,564 | 35.540 | 15.21 | 8.599 | 74.890 |
| Direct | 41,564 | 8.695 | 1.808 | 0.000 | 19.000 |
| Ind | 41,564 | 4.780 | 3.279 | 1.000 | 18.000 |
| Act | 41,564 | 0.397 | 0.489 | 0.000 | 1.000 |
| ProvNum | 41,564 | 0.831 | 0.375 | 0.000 | 1.000 |
| GDP | 41,564 | 43485.370 | 32818.510 | 189.090 | 129118.600 |

Since the same violation of a company may be punished or announced by different departments, duplicate penalty announcements of the same violation are manually removed, leaving one penalty announcement record for each violation of a company. Data on other variables of listed companies are sourced both from RESSET and CSMAR database. All continuous variables are winsorized at the 1% level, and that results in a total of 4,511 companies with 41,564 observations.

## Empirical results analysis

**China's listed company survival risk KM estimation.** The non-parametric KM estimation is used to plot the intuitive survival function and hazard function graphs of companies, which analyzes the impact of company misconduct on the cumulative survival risk and survival rate of company. V = 0 indicates the company has no misconduct during its survival period, and V = 1 indicates the company has engaged in misconduct during its survival period.

As shown in Fig 1 Hazard function curve and Fig 2 Survival function curve, the horizontal axis represents the survival time (years), while the vertical axis in (a) represents the hazard value and the vertical axis in (b) represents the survival value. From the graph, it can be visually observed that as time increases, the survival risk of company gradually increases, and the cumulative survival rate gradually decreases. The hazard curve of companies with misconduct is above that of companies without misconduct, and the cumulative survival curve of companies with misconduct is below that of companies without misconduct. This implies that compared to companies without misconduct, companies with misconduct have higher survival risk and lower survival rate. Preliminary conclusion can be drawn from Figs 1 and 2: without considering other factors, misconduct significantly reduce the survival rate of company. However, there are many factors that can affect company survival risk. In order to accurately describe the relationship between company misconduct and company survival risk, the Cloglog survival model is employed for a more rigorous estimation.

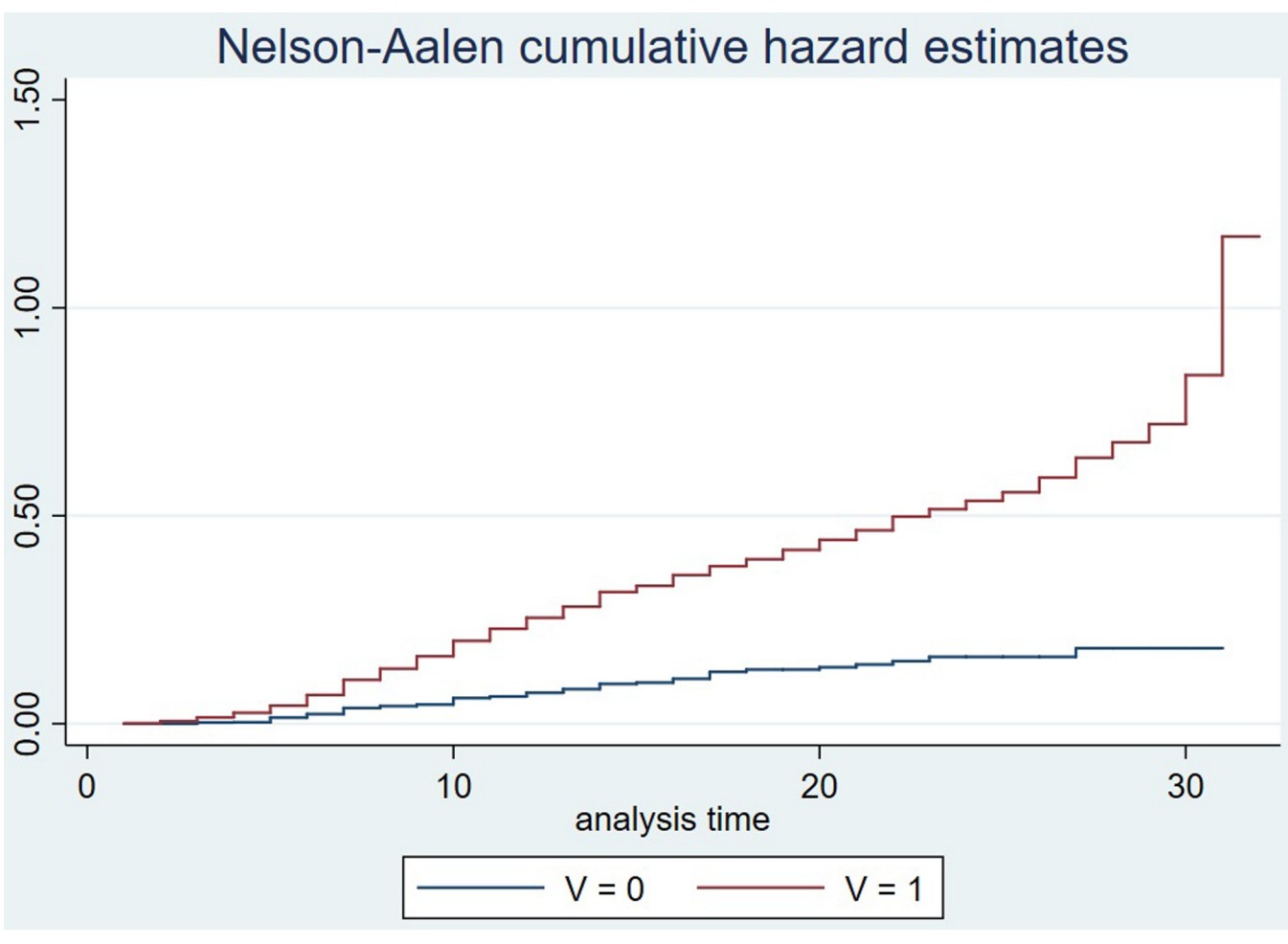

**Fig 1.**

**Survival estimation based on the Clog-log survival model.** Table 3 reports the regression results of the Clog-log survival model in Eq (3), which examines whether misconduct by companies affect their survival risk. Columns (1) presents the estimation results with only the inclusion of misconduct by company, while column (2) presents the estimation results with the inclusion of control variables, and column (3) further controls for time and industry effects. It can be observed from Table 3 that the regression coefficients of misconduct are positive and significant at the 1% level in columns (1)-(3). According to the previous description of the model, positive regression coefficients imply that $e^{\beta_1} > 1$, which indicates that the variable increases the company's survival risk. Taking the most comprehensive regression result in column (3) as an example, the coefficient of the core independent variable(*Violate*) is 0.9462. This indicates that, keeping other factors constant, the survival risk of company with misconduct is 157.59% ($= e^{0.9462} - 1$)higher than that of company without misconduct, strongly supporting hypothesis H0. Additionally, the coefficients of the control variables align with theoretical expectations. Variables such as total assets (*Asset*), company age (*Age*), total asset turnover ratio (*Assetturn*), return on total assets (*Return*), free cash flow (*FC*), asset growth rate (*Assetgrow*), Tobin's Q (*Q*), company board size (*Director*) are negative and significant at the 1% level, which means that these are protective factors for the survival of company. Ownership(*Act*) and company location

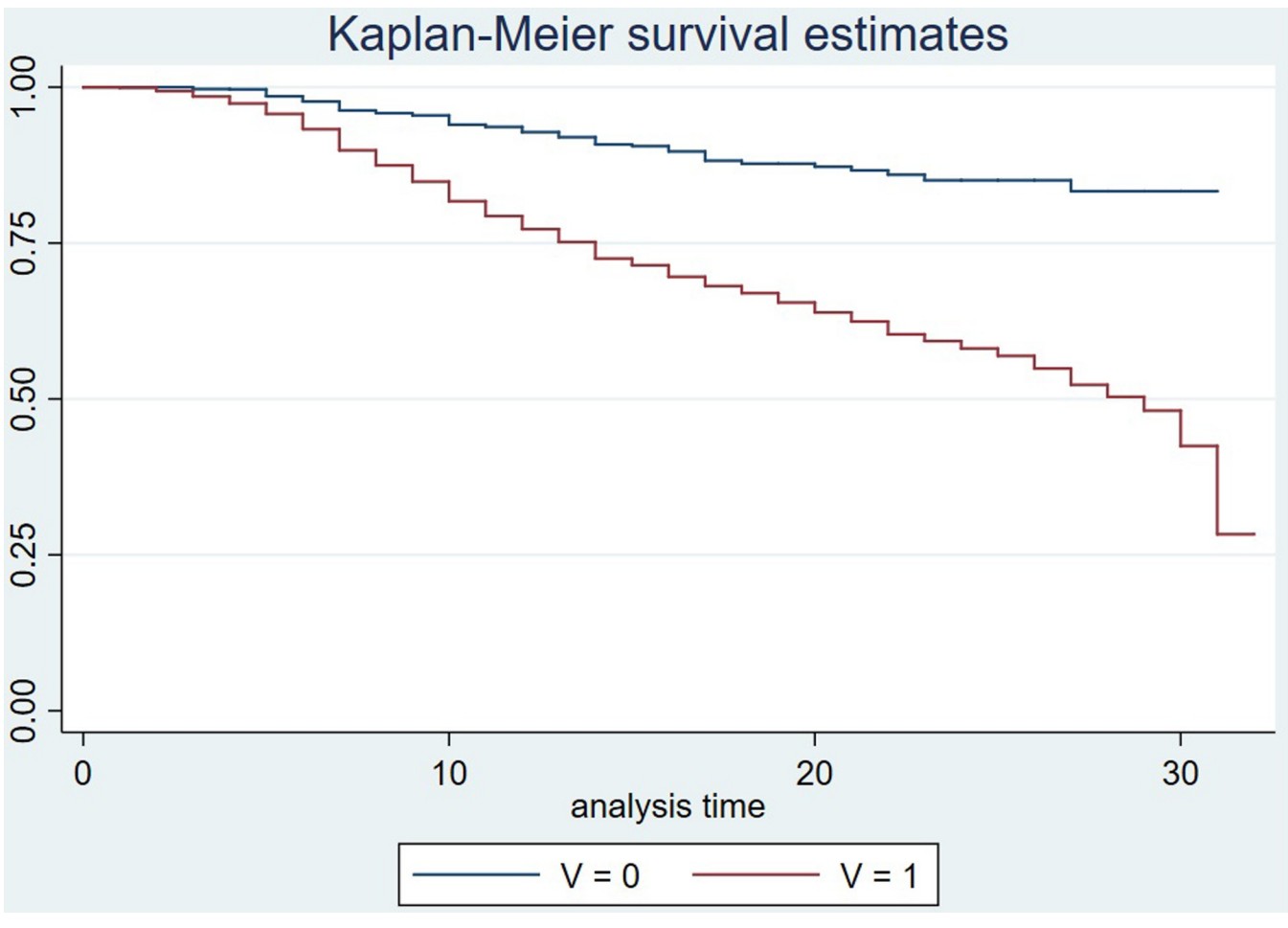

**Fig 2.**

(*ProvNum*) are also negative and significant, which means that the state-owned company and company located in the eastern region have a higher survival rate.

## Impact mechanisms analysis

The main research findings in the previous sections indicate that company misconduct leads to an increase in the risk of company survival. According to the theoretical analysis in the previous sections, company misconduct may result in a decline in corporate value, damage to reputation, difficulties in financing, and a decrease in investor confidence. In this study, we focus on investor confidence and financing constraints to examine impact mechanisms. Drawing on the research by Zhao et al, Wen and Ye [90, 91], a two-stage mediation effect model is used:

$$M_{it} = \alpha + aViolate_{it} + \gamma Controls_{it} + \lambda_t + \eta_i + \varepsilon_{it} \tag{4}$$

$$F(h_{it}) = \alpha + cViolate_{it} + bM_{it} + \gamma Controls_{it} + \lambda_t + \eta_i + \varepsilon_{it} \tag{5}$$

$M_{it}$ represents the mechanisms of how company misconduct affects its survival risk, which includes the reputation mechanism and the financing constraint mechanism;

**Table 3. Basic regression results of company violations and company survival risk.**

| Variables | (1) | (2) | (3) |
|---|---|---|---|
| Violate | 1.2898*** | 0.9651*** | 0.9462*** |
|  | (148.35) | (108.65) | (106.01) |
| Asset |  | -0.1969*** | -0.2976*** |
|  |  | (-79.92) | (-99.45) |
| Age |  | 0.0604*** | 0.0635*** |
|  |  | (109.30) | (111.74) |
| FA |  | 0.0061*** | -0.0071*** |
|  |  | (3.75) | (-4.05) |
| Assetturn |  | -0.3822*** | -0.2674*** |
|  |  | (-65.58) | (-41.91) |
| Return |  | -0.8329*** | -0.4352*** |
|  |  | (-13.40) | (-6.83) |
| FL |  | 0.0726*** | 0.0838*** |
|  |  | (36.19) | (40.72) |
| FC |  | -0.7873*** | -0.8039*** |
|  |  | (-36.19) | (-36.61) |
| Assetgrow |  | -0.1464*** | -0.0785*** |
|  |  | (-16.79) | (-8.90) |
| Q |  | 0.0623*** | 0.0254*** |
|  |  | (17.98) | (6.94) |
| PB |  | 0.0313*** | 0.0312*** |
|  |  | (23.24) | (21.90) |
| Debtass |  | 1.9038*** | 1.9677*** |
|  |  | (94.73) | (90.70) |
| Owncon |  | 0.0026*** | 0.0036*** |
|  |  | (15.43) | (20.60) |
| Director |  | -0.0650*** | -0.0478*** |
|  |  | (-47.18) | (-33.65) |
| Act |  | -0.2839*** | -0.2728*** |
|  |  | (-54.16) | (-50.72) |
| ProvNum |  | -0.2112*** | -0.1220*** |
|  |  | (-35.48) | (-19.60) |
| GDP |  | -0.0000*** | -0.0000*** |
|  |  | (-59.56) | (-83.23) |
| _cons | -2.5951*** | 0.6788*** | 2.4841*** |
|  | (-285.82) | (12.41) | (38.91) |
| Year | NO | NO | YES |
| Ind | NO | NO | YES |
| N | 41,564 | 41,564 | 41,564 |
| Log likelihood | -371636.71 | -333043.42 | -326574.41 |

Note:*,**and***represent significance levels at 0.1, 0.05 and 0.01, respectively. Standard errors are shown in parentheses.

$F(h_{it})$ represents the dependent variable, which is company survival risk. We conduct mediation effect tests in the following way: under the condition of significant coefficient $\beta_1$ in the previous model (3), models (4) and (5) are tested. If the coefficients a, b and c are significant, and the signs of a×b and c are the same, and there is a partial mediation effect; if the

coefficients a and b are significant, the coefficient c is not significant, and signs of a×b and c are the same, and there is a complete mediation effect; finally, conduct a bootstrap test. The results of mechanism analysis are shown in Table 4.

## Investor confidence mechanism

According to the analysis in the previous sections, when a company is punished or regulated due to misconduct, it may face public criticisms, investigations, fines, rectifications, and in severe cases, the responsible individuals of the company may be criminally detained, which directly affects the normal business operations of the company. At the same time, negative news about company misconduct can spread in the capital market, which leads to damage to the company's reputation and undermining investor confidence, and causes concern and speculation about the company among the capital market and investors. Consequently, this affects the company's market value, stock price, financing ability, etc., and increases the survival risk of the company. This paper takes the annual turnover rate (Annual Stock Trading Volume / Outstanding Shares) of company stocks as a proxy variable for investor confidence (*Confin*), with a higher value indicating higher investor confidence in the company. Column (1) in Table 4 shows that company misconduct has a negative estimated coefficient on the investor confidence and significant at the 1% level. Column (2) in Table 4 indicates that the estimated coefficient of the investor confidence index on the company survival risk is negative, while the estimated coefficient of company misconduct on company survival risk is positive, and all of these coefficients are significant at the 1% level. The bootstrap test with confidence interval excluding 0 and the partial mediation effect test is passed, which indicates that company misconduct increases its survival risk by reducing investor confidence. The above conclusions effectively validate hypothesis H1.

## Financing constraint mechanism

According to the theoretical analysis in the previous sections, company misconduct will increase the degree of financing constraint. This paper respectively uses SA index [92] and KZ index [93] to measure the degree of financing constraint for company, where a higher SA and KZ index indicates a more severe financing constraint for companies. The results in columns (3) and (5) of Table 4 show that the estimated coefficients of company misconduct (*Violate*) on financing constraint KZ and SA index are positive and significant at the 1% level. The results in columns (4) and (6) of Table 4 show that the estimated coefficients of corporate misconduct (*Violate*), and financing constraint KZ and SA index on the company survival risk are all positive and significant at the 1% level. The bootstrap test with financing constraint intervals excluding 0, which indicates that the partial mediation effect test is passed. These results indicate company misconduct increases company survival risk by increasing the degree of financing constraints. These conclusions effectively validate hypothesis H2.

## Endogeneity treatment and robustness tests

**Endogeneity treatment.** The estimation results in Table 3 provide strong evidence for the relationship between company misconduct and the risk of company survival. However, there may be endogeneity issues caused by reverse causality between misconduct and company survival risk. In other word, the greater the company survival risk, the more likely it is for company to embellish their performance through fraudulent disclosure, stock price manipulation, and misappropriation of corporate assets, thereby conceal potential risks and delay the "explosion" time. Although this study attempts to control for various factors that influence company survival as much as possible, the empirical results may still be affected by some unobservable

**Table 4. Examination of impact mechanisms.**

| Variables | Investor Confidence Mechanism | | Financing Constraint Mechanism | | | |
| --- | --- | --- | --- | --- | --- | --- |
| | (1)Investor Confidence Mechanism(Confin) | (2)survival risk | (3)KZ Index | (4)survival risk | (5)SA Index | (6)survival risk |
| Violate | -53.4002*** | 1.3909*** | 0.2375*** | 1.2644*** | 0.0572*** | 1.3070*** |
| | (-10.03) | (30.73) | (15.35) | (25.90) | (20.22) | (26.85) |
| KZ | | | | 0.1623*** | | |
| | | | | (11.76) | | |
| SA | | | | | | 1.6222*** |
| | | | | | | (21.28) |
| Confin | | -0.0003*** | | | | |
| | | (-7.75) | | | | |
| Asset | -136.4312 *** | -0.2931*** | -0.0913*** | -0.2034*** | | |
| | (-58.74) | (-16.59) | (-11.89) | (-11.04) | | |
| Age | -6.5391*** | 0.0788*** | 0.0017 | 0.0869*** | | |
| | (-13.43) | (26.90) | (1.23) | (27.20) | | |
| FA | 3.5890*** | -0.0122 | -0.1058*** | 0.0128 | -0.0084*** | 0.0017 |
| | (2.62) | (-1.38) | (-21.19) | (1.26) | (-11.40) | (0.16) |
| Assetturn | 27.5124*** | -0.3638*** | -0.2430*** | -0.2534*** | 0.0089*** | -0.2765*** |
| | (5.17) | (-9.23) | (-12.93) | (-6.18) | (3.08) | (-6.62) |
| Return | -17.8522 | -0.9188** | -14.0018*** | 0.2265 | 0.2638*** | -3.2103*** |
| | (-0.33) | (-2.31) | (-58.55) | (0.48) | (9.71) | (-7.31) |
| FL | -3.3331* | 0.1334*** | 0.0286*** | 0.1200*** | 0.0057*** | 0.1209*** |
| | (-1.70) | (9.87) | (4.98) | (8.35) | (4.76) | (8.60) |
| FC | -1.2e+02*** | -0.9811*** | -0.2321*** | -0.8738*** | 0.0051 | -0.9308*** |
| | (-5.77) | (-7.58) | (-2.63) | (-5.59) | (0.43) | (-6.00) |
| Assetgrow | 44.1026*** | -0.2359*** | -1.1440*** | 0.2614*** | -0.0247*** | -0.0568 |
| | (5.29) | (-4.44) | (-27.30) | (4.33) | (-5.29) | (-0.96) |
| Q | -1.4e+02*** | 0.0542*** | 0.4584*** | -0.0724*** | -0.0013 | 0.0397* |
| | (-32.35) | (2.75) | (32.46) | (-2.87) | (-0.67) | (1.65) |
| PB | 42.7479*** | 0.0363*** | 0.0092 | 0.0484*** | -0.0049*** | 0.0791*** |
| | (21.32) | (4.51) | (1.43) | (4.65) | (-5.77) | (7.99) |
| Debtass | -24.1262 | 2.4592*** | 5.9432*** | 0.9760*** | -0.0316*** | 1.4447*** |
| | (-1.28) | (19.11) | (95.31) | (6.03) | (-3.18) | (11.54) |
| Owncon | -0.0413 | 0.0009 | -0.0068*** | 0.0043*** | -0.0023*** | 0.0011 |
| | (-0.26) | (0.90) | (-14.25) | (4.00) | (-26.72) | (1.08) |
| Director | -2.1358* | -0.0706*** | -0.0297*** | -0.0558*** | -0.0015** | -0.0791*** |
| | (-1.88) | (-8.41) | (-8.05) | (-6.21) | (-2.18) | (-9.06) |
| Act | -14.6368*** | -0.1727*** | 0.0708*** | -0.2386*** | 0.0548*** | -0.2544*** |
| | (-3.10) | (-5.40) | (4.78) | (-6.95) | (20.92) | (-7.51) |
| ProvNum | -25.6431*** | -0.2222*** | 0.0499*** | -0.2376*** | 0.0040 | -0.2532*** |
| | (-4.30) | (-6.01) | (2.76) | (-6.00) | (1.31) | (-6.45) |
| GDP | 0.0004*** | -0.0000*** | -0.0000*** | -0.0000*** | 0.0000 | -0.0000*** |
| | (3.80) | (-15.67) | (-4.49) | (-15.18) | (0.94) | (-15.04) |
| _cons | 3561.279*** | 2.6242*** | 4.6054*** | 0.0466 | 1.9988*** | -6.5211*** |
| | (68.22) | (7.00) | (27.16) | (0.12) | (43.41) | (-20.35) |
| Year | YES | YES | YES | YES | YES | YES |
| Ind | YES | YES | YES | YES | YES | YES |
| N | 40,749 | 40,749 | 34,625 | 34,625 | 41,564 | 41,564 |

(*Continued*)

**Table 4.** (Continued)

| Variables | Investor Confidence Mechanism | | Financing Constraint Mechanism | | | |
|---|---|---|---|---|---|---|
| | (1)Investor Confidence Mechanism(Confin) | (2)survival risk | (3)KZ Index | (4)survival risk | (5)SA Index | (6)survival risk |
| BootstrapTests | [0.0011074,0.0021382] | | [0.0022032,0.0037648 ] | | [0.0021952,0.0039588] | |

Note: in order to avoid the influence of multicollinearity, the variable "Assets" and "Age" are excluded from the control variables in column (3) as it is included in the construction of SA index; *,**and***represent significance levels at 0.1, 0.05 and 0.01, respectively.

factors, that is omitted variables. To mitigate the above endogeneity issues, the following approaches are adopted in this study.

**Lagged regression.**   In this part, the core independent variable "*Violate*" is lagged by one, two, and three periods, and the Clog-log survival model is used for testing. Table 5 reports the regression results of the lagged independent variables. From Table 5, it can be seen that there are positive correlations between company misconduct and the survival risk in the next one to three periods. These correlations are significant at the 1% level, which indicates that company misconduct not only increases the current survival risk of the company but also increases the future survival risk.

**Propensity score matching (PSM).**   In order to mitigate the endogeneity problem caused by sample self-selection, propensity score matching (PSM) is used in this study. The same sample is still used as the research subjects, and the sample is divided into companies with misconduct and without misconduct. The companies with misconduct are the treatment group, while the companies without misconduct are the control group. By matching as much as possible to eliminate selection bias, a more accurate effect evaluation can be obtained. The specific procedures are as follows: (1) the survival of the company (whether it has been specially treated) is the dependent variable; (2) the company misconduct (*Violate*) is the independent variable, with a value of 1 for company with misconduct and is in the treatment group, value of 0 for company without misconduct and is in the control group; (3) use all control variables in the survival analysis model as PSM covariates, and match using nearest neighbor matching(n = 1, n = 4), radius matching and local linear regression matching methods, respectively; (4) calculate the average treatment effect on the Treated (ATT). Table 6 reports the empirical analysis results of the ATT of company misconduct on survival risk under different matching methods. From Table 6, it can be seen that regardless of the matching method used, the ATT of the independent variable is positive and significant at the 1% level. This indicates that even after controlling for eliminate sample selection bias through matching, the misconduct of listed company still leads to an increased risk of survival, which provides strong support for the previous research conclusion.

**Robustness test.**   In order to enhance the reliability of the conclusions, this study adopts robustness tests by replacing the dependent variables and estimation models. The results are shown in Table 7.

**Replace the dependent variable.**   In the discussion of the impact of company misconduct on the risk of survival, this study considers the endpoint event for survival analysis as the company is specially treated (including the first time being ST or *ST) for the first time. In order to test the robustness of the results, this study uses the bankruptcy warning index O-Score [94] as a proxy variable for survival risk. This index reflects the likelihood of a company going bankrupt, with a higher value indicating a greater risk of bankruptcy. If the index is greater than 0.5, it indicates a high risk of bankruptcy for the company. The regression results reported in column (1) of Table 7, which shows that the impact of misconduct on the company survival risk is positive and significant, confirming the robustness of the previous regression results.

**Table 5. Lagged regression of company misconduct and survival risk.**

| Variables | (1) lagged 1 period | (2) lagged 2 period | (3) lagged 3 period |
|---|---|---|---|
| Violate | 0.9425*** | 0.9308*** | 0.8656*** |
| | (94.75) | (88.13) | (80.37) |
| Asset | -0.2794*** | -0.2648*** | -0.2928*** |
| | (-82.10) | (-74.51) | (-79.69) |
| Age | 0.0694*** | 0.0710*** | 0.0714*** |
| | (108.04) | (104.77) | (100.81) |
| FA | 0.0104*** | 0.0253*** | 0.0341*** |
| | (5.21) | (12.05) | (15.36) |
| Assetturn | -0.2163*** | -0.2239*** | -0.2733*** |
| | (-29.99) | (-29.28) | (-34.16) |
| Return | -0.8551*** | -1.2408*** | -0.3451*** |
| | (-11.23) | (-15.48) | (-4.34) |
| FL | 0.1001*** | 0.0872*** | 0.0976*** |
| | (39.03) | (34.57) | (38.69) |
| FC | -0.6990*** | -0.7565*** | -0.8529*** |
| | (-24.17) | (-25.71) | (-29.03) |
| Assetgrow | 0.2047*** | 0.1955*** | 0.0140 |
| | (18.44) | (17.31) | (1.17) |
| Q | -0.0068 | -0.0090* | -0.0336*** |
| | (-1.44) | (-1.84) | (-6.81) |
| PB | 0.0326*** | 0.0365*** | 0.0382*** |
| | (17.01) | (18.02) | (19.54) |
| Debtass | 1.8448*** | 1.8084*** | 1.9826*** |
| | (72.63) | (67.92) | (71.72) |
| Owncon | 0.0048*** | 0.0043*** | 0.0043*** |
| | (23.69) | (20.09) | (19.11) |
| Director | -0.0534*** | -0.0598*** | -0.0569*** |
| | (-32.57) | (-33.84) | (-31.02) |
| Act | -0.3018*** | -0.3073*** | -0.3170*** |
| | (-49.36) | (-47.76) | (-47.54) |
| ProvNum | -0.1447*** | -0.1339*** | -0.1119*** |
| | (-20.27) | (-17.89) | (-14.36) |
| GDP | -0.0000*** | -0.0000*** | -0.0000*** |
| | (-76.60) | (-76.98) | (-74.02) |
| _cons | 1.8192*** | 1.5116*** | 2.0019*** |
| | (25.03) | (19.90) | (25.21) |
| Year | YES | YES | YES |
| Ind | YES | YES | YES |
| N | 33,943 | 30,027 | 26,697 |
| Log likelihood | -265151.29 | -242428.99 | -223795.74 |

Note:*,**and***represent significance levels at 0.1,0.05 and 0.01,respectively. Standard errors are shown in parentheses.

**Change estimation models.** This study uses discrete time logit and probit models, cox proportional hazard model, as well as ordinary logit and probit models for estimation. Although the estimated coefficients of each model are not comparable, if the estimated coefficients and significance levels of each model are consistent with the previous results, it can

**Table 6. Empirical analysis results of the impact of company misconduct on survival risk under different matching methods.**

| PSM<br><br>Average Treatment Effect(ATT) | Nearest neighbor(n = 1) | Nearest neighbor(n = 4) | Radius<br>(cal = 0.01) | Local Linear Regression<br>(bandwidth = 0.8) |
|---|---|---|---|---|
| Average Treatment Effect(ATT) | 0.1521*** | 0.1511*** | 0.1505*** | 0.1424*** |
| Standard Errors | 0.0049 | 0.0040 | 0.0038 | 0.0049 |
| T-stat | 30.98 | 36.89 | 39.39 | 29.01 |

Note:*,**and***represent significance levels at 0.1,0.05 and 0.01,respectively.

mostly validate the robustness of the previous results. The regression results in columns (2) to (6) of Table 7 further verify the relationship between company misconduct and company survival.

**Heterogeneity analysis.** The impact of company misconduct on company survival may vary due to heterogeneity. Subsequently, this paper adopts grouped regression and full-sample interaction regression to examine heterogeneity of ownership, size, governance, and regional factors. The results of heterogeneity tests are show in Table 8.

**Ownership heterogeneity.** This paper divides the sample into state-owned and non-state-owned company based on ownership heterogeneity, and the results of grouped regression and full-sample interaction regression are shown in Table 8, columns (1)-(3). The results indicate that the coefficient of misconduct (*Violate*) in the state-owned company group is significantly lower than that in the non-state-owned company group, and the coefficient of the interaction of company misconduct and ownership dummy variables (*Violate*Act*) in the regression is -0.9649, significant at the 1% level. These indicate that after misconduct, state-owned company have a lower survival risk than non-state-owned company.

**Size heterogeneity.** In this paper, companies are classified as large-size and small-size based on the average size within the industry for the year. Companies with a size higher than the industry average for that year are considered as large-size group, while those with a size lower than the industry average are considered as small -size group. Columns (4) and (5) in Table 8 report the results of regression of large-size and small-size company, and column (6) reports the results of full-sample interaction regression. The results indicate that the coefficient of large-size company misconduct (*Violate*) is significantly lower than of small-size company. The coefficient of the interaction of company misconduct and company size dummy variable (*Violate*Asset*) is -0.0194 and significant at the 1% level, suggesting that after misconduct, the survival risk of large-size company is lower than that of small-size company.

**Governance heterogeneity.** This paper calculates the annual average shareholding ratio of the largest shareholder of listed companies. Companies with a higher shareholding ratio than the average are considered to have a high equity concentration, while those with a lower ratio than the average are considered to have a low equity concentration. Based on the heterogeneity of equity concentration, grouping and interaction tests are showed in Table 8, columns (7)-(9). The results indicate that the coefficient of corporate misconduct (*Violate*) in high equity concentration company is significantly lower than that in low equity concentration company, the coefficient of the interaction of corporate misconduct and equity concentration dummy variable (*Violate*Owncon*) is -0.0049 and significant at the 1% level. These results suggest that after misconduct, high equity concentration company have a lower survival risk compared to low equity concentration company.

**Regional heterogeneity.** This paper divides the sample companies into eastern and central-western regions based on their locations. Table 8 reports the results of regression for the

Table 7. Robustness test.

| Variables | Replace the Dependent Variable | Change Estimation Models | | | | |
|---|---|---|---|---|---|---|
| | (1)Bankruptcy Warning Index | (2)discrete time Logit | (3)discrete time Probit | (4)Cox | (5)Logit | (6)Probit |
| Violate | 0.0715*** | 1.0908*** | 0.6157*** | 1.0020*** | 1.4372*** | 0.7789*** |
| | (7.27) | (110.50) | (116.36) | (23.68) | (31.56) | (34.32) |
| Asset | | -0.3684*** | -0.2053*** | -0.3540*** | -0.2606*** | -0.1398*** |
| | | (-98.44) | (-96.48) | (-25.55) | (-15.26) | (-14.73) |
| Age | -0.0004 | 0.0815*** | 0.0479*** | 0.0136*** | 0.0811*** | 0.0470*** |
| | (-0.47) | (111.16) | (112.89) | (5.31) | (27.77) | (28.50) |
| FA | -0.2353*** | -0.0090*** | -0.0062*** | -0.0084 | -0.0149* | -0.0082* |
| | (-35.11) | (-4.36) | (-5.21) | (-1.13) | (-1.69) | (-1.69) |
| Assetturn | 0.0593*** | -0.3031*** | -0.1717*** | -0.2714*** | -0.3638*** | -0.2032*** |
| | (5.12) | (-38.86) | (-38.56) | (-9.20) | (-9.21) | (-9.34) |
| Return | -16.4334*** | -0.6449*** | -0.4278*** | -0.4216 | -0.9169** | -0.5522** |
| | (-116.31) | (-8.30) | (-9.71) | (-1.53) | (-2.32) | (-2.57) |
| FL | -0.0040 | 0.1339*** | 0.0796*** | 0.0710*** | 0.1344*** | 0.0810*** |
| | (-1.02) | (46.03) | (46.01) | (7.72) | (9.95) | (10.26) |
| FC | 0.0178 | -0.9715*** | -0.5698*** | -0.6917*** | -0.9820*** | -0.5451*** |
| | (0.33) | (-34.51) | (-34.70) | (-7.63) | (-7.55) | (-7.46) |
| Assetgrow | -0.1072*** | -0.0991*** | -0.0625*** | 0.0429 | -0.2549*** | -0.1467*** |
| | (-6.07) | (-9.03) | (-9.88) | (1.18) | (-4.76) | (-4.99) |
| Q | -0.0507*** | 0.0100** | 0.0022 | -0.0082 | 0.0890*** | 0.0472*** |
| | (-5.34) | (2.05) | (0.78) | (-0.55) | (4.68) | (4.31) |
| PB | 0.0798*** | 0.0530*** | 0.0338*** | 0.0295*** | 0.0258*** | 0.0166*** |
| | (20.35) | (26.21) | (28.51) | (5.40) | (3.29) | (3.69) |
| Debtass | 6.1309*** | 2.3773*** | 1.3572*** | 1.7465*** | 2.4764*** | 1.3761*** |
| | (140.57) | (88.23) | (87.29) | (18.37) | (19.20) | (19.12) |
| Owncon | -0.0046*** | 0.0047*** | 0.0026*** | 0.0063*** | 0.0012 | 0.0004 |
| | (-15.69) | (21.13) | (19.91) | (7.79) | (1.17) | (0.69) |
| Director | -0.0489*** | -0.0603*** | -0.0352*** | -0.0526*** | -0.0707*** | -0.0408*** |
| | (-21.13) | (-33.63) | (-33.91) | (-8.11) | (-8.40) | (-8.65) |
| Act | -0.1782*** | -0.3488*** | -0.1937*** | -0.3125*** | -0.1704*** | -0.0792*** |
| | (-19.32) | (-50.93) | (-48.29) | (-12.64) | (-5.31) | (-4.34) |
| ProvNum | -0.0679*** | -0.2178*** | -0.1385*** | -0.1536*** | -0.2134*** | -0.1363*** |
| | (-5.97) | (-26.99) | (-29.12) | (-5.39) | (-5.78) | (-6.42) |
| GDP | 0.0000*** | -0.0000*** | -0.0000*** | -0.0000*** | -0.0000*** | -0.0000*** |
| | (3.25) | (-78.88) | (-77.10) | (-12.93) | (-16.11) | (-15.71) |
| _cons | -8.7623*** | 3.7190*** | 1.9988*** | | 1.7035*** | 0.8317*** |
| | (-139.39) | (46.38) | (43.41) | | (4.78) | (4.15) |
| Year | YES | YES | YES | YES | YES | YES |
| Industry | YES | YES | YES | YES | YES | YES |
| N | 34,625 | 41,564 | 41,564 | 41,564 | 41,564 | 41,564 |

Note:in order to avoid the influence of multi-collinearity, the variable "Assets" are excluded from the control variables in column (1) as it is included in the construction of O-Score; *,**and***represent significance levels at 0.1, 0.05 and 0.01, respectively.

two sub-samples in columns (10) and (11), and column (12) reports the results of interaction regression for the full sample. The results show that the coefficient of company misconduct (*Violate*) in eastern region is significantly higher than that in central-western region, the

**Table 8. Heterogeneity test.**

| Variables | Ownership Heterogeneity | | | Size Heterogeneity | | | Governance Heterogeneity | | | Regional Heterogeneity | | |
|---|---|---|---|---|---|---|---|---|---|---|---|---|
| | (1) State-owned | (2) Non-state-owned | (3) Full Sample | (4) Large-size | (5) Small-size | (6) Full Sample | (7) High equity concentration | (8) Low equity concentration | (9) Full Sample | (10) Eastern | (11) Central-western | (12) Full Sample |
| Violate | 0.6460*** | 1.1549*** | 1.5865*** | 0.7283*** | 1.1902*** | 0.9559*** | 0.8538*** | 1.1256*** | 1.1403*** | 1.0465*** | 0.7041*** | 0.5847*** |
| | (61.09) | (79.21) | (88.79) | (61.91) | (85.11) | (98.80) | (72.68) | (79.77) | (48.47) | (99.97) | (40.10) | (33.88) |
| Violate*Act | | | -0.9649*** | | | | | | | | | |
| | | | (-46.74) | | | | | | | | | |
| Violate*Asset | | | | | | -0.0194*** | | | | | | |
| | | | | | | (-2.59) | | | | | | |
| Violate*Owncon | | | | | | | | | -0.0049*** | | | |
| | | | | | | | | | (-9.03) | | | |
| Violate*ProvNum | | | | | | | | | | | | 0.4691*** |
| | | | | | | | | | | | | (23.39) |
| Asset | -0.3862*** | -0.2391*** | -0.3057*** | | | -0.2915*** | -0.3224*** | -0.3016*** | -0.2981*** | -0.3093*** | -0.2872*** | -0.2973*** |
| | (-91.32) | (-54.72) | (-101.98) | | | (-76.79) | (-72.28) | (-73.25) | (-99.61) | (-90.13) | (-45.29) | (-99.46) |
| Age | 0.0457*** | 0.0769*** | 0.0636*** | 0.0605*** | 0.0608*** | 0.0635*** | 0.0621*** | 0.0688*** | 0.0636*** | 0.0627*** | 0.0863*** | 0.0637*** |
| | (52.13) | (98.38) | (111.68) | (70.45) | (78.15) | (111.78) | (69.04) | (91.79) | (111.95) | (95.13) | (68.20) | (112.02) |
| FA | 0.0297*** | -0.0299*** | -0.0024 | 0.0119*** | -0.0104*** | -0.0070*** | -0.0014 | -0.0118*** | -0.0065*** | -0.0101*** | 0.0027 | -0.0079*** |
| | (9.71) | (-13.25) | (-1.39) | (2.81) | (-5.20) | (-4.01) | (-0.46) | (-5.50) | (-3.68) | (-4.76) | (0.83) | (-4.51) |
| Assetturn | -0.1529*** | -0.4333*** | -0.2656*** | -0.2504*** | -0.2612*** | -0.2669*** | -0.2367*** | -0.3317*** | -0.2672*** | -0.2142*** | -0.4317*** | -0.2681*** |
| | (-17.80) | (-44.33) | (-41.67) | (-27.25) | (-29.31) | (-41.80) | (-25.22) | (-37.77) | (-41.86) | (-29.48) | (-30.48) | (-42.04) |
| Return | 0.8925*** | -0.9450*** | -0.3424*** | 0.2967*** | -1.1863*** | -0.4310*** | 0.4437*** | -0.4143*** | -0.4302*** | -0.5110*** | 0.0568 | -0.4272*** |
| | (9.41) | (-10.73) | (-5.37) | (2.83) | (-14.69) | (-6.76) | (4.30) | (-5.01) | (-6.75) | (-6.77) | (0.46) | (-6.70) |
| FL | 0.1066*** | 0.0570*** | 0.0854*** | 0.1142*** | 0.0618*** | 0.0839*** | 0.0803*** | 0.0852*** | 0.0836*** | 0.0946*** | 0.0645*** | 0.0837*** |
| | (38.23) | (18.16) | (41.46) | (40.73) | (20.27) | (40.74) | (23.62) | (32.73) | (40.62) | (39.78) | (14.98) | (40.68) |
| FC | -0.7305*** | -0.8316*** | -0.7885*** | -0.5719*** | -1.0827*** | -0.8032*** | -0.5479*** | -0.9401*** | -0.8028*** | -0.7480*** | -0.8517*** | -0.8089*** |
| | (-21.09) | (-28.57) | (-35.88) | (-15.22) | (-38.65) | (-36.58) | (-15.17) | (-33.56) | (-36.56) | (-28.71) | (-20.03) | (-36.86) |
| Assetgrow | -0.0661*** | -0.0716*** | -0.0703*** | 0.1002*** | -0.2866*** | -0.0780*** | -0.0860*** | -0.0585*** | -0.0777*** | -0.0486*** | -0.1524*** | -0.0785*** |
| | (-4.52) | (-6.37) | (-7.98) | (7.99) | (-23.29) | (-8.84) | (-5.85) | (-5.27) | (-8.81) | (-4.80) | (-8.23) | (-8.90) |
| Q | -0.0573*** | 0.0351*** | 0.0198*** | -0.2853*** | 0.1105*** | 0.0258*** | -0.1197*** | 0.0727*** | 0.0255*** | 0.0119*** | 0.0250*** | 0.0236*** |
| | (-9.57) | (7.30) | (5.39) | (-34.35) | (26.48) | (7.06) | (-17.18) | (16.22) | (6.97) | (2.73) | (3.55) | (6.46) |
| PB | 0.0559*** | 0.0291*** | 0.0320*** | 0.0876*** | 0.0587*** | 0.0311*** | 0.0391*** | 0.0306*** | 0.0310*** | 0.0430*** | 0.0166*** | 0.0318*** |
| | (25.10) | (15.01) | (22.43) | (33.45) | (35.53) | (21.86) | (15.43) | (17.01) | (21.74) | (25.75) | (5.85) | (22.34) |
| Debtass | 2.2120*** | 1.6198*** | 1.9868*** | 1.7403*** | 1.2540*** | 1.9697*** | 2.3391*** | 1.7366*** | 1.9733*** | 2.2120*** | 1.2067*** | 1.9625*** |
| | (69.48) | (52.10) | (91.49) | (53.80) | (44.41) | (90.72) | (67.46) | (60.74) | (90.94) | (87.83) | (26.71) | (90.43) |
| Owncon | 0.0075*** | -0.0015*** | 0.0033*** | 0.0037*** | 0.0006** | 0.0036*** | | | 0.0081*** | 0.0035*** | 0.0054*** | 0.0037*** |
| | (29.52) | (-5.97) | (18.79) | (15.08) | (2.27) | (20.49) | | | (15.42) | (17.31) | (14.58) | (21.21) |
| Director | -0.0579*** | -0.0306*** | -0.0472*** | -0.0636*** | -0.0582*** | -0.0478*** | -0.0743*** | -0.0275*** | -0.0479*** | -0.0307*** | -0.0824*** | -0.0477*** |
| | (-29.25) | (-14.33) | (-33.23) | (-33.27) | (-28.06) | (-33.67) | (-35.48) | (-14.37) | (-33.77) | (-18.27) | (-28.47) | (-33.59) |
| Act | | | 0.6317*** | -0.4285*** | -0.2378*** | -0.2728*** | -0.0952*** | -0.3781*** | -0.2738*** | -0.3269*** | -0.0346*** | -0.2703*** |
| | | | (31.14) | (-53.45) | (-32.54) | (-50.73) | (-10.96) | (-53.75) | (-50.88) | (-51.54) | (-3.21) | (-50.28) |
| ProvNum | -0.1845*** | -0.1142*** | -0.1345*** | -0.1485*** | -0.0712*** | -0.1216*** | -0.1243*** | -0.1104*** | -0.1244*** | | | -0.5553*** |
| | (-21.18) | (-12.38) | (-21.56) | (-15.42) | (-8.57) | (-19.53) | (-12.85) | (-13.34) | (-19.96) | | | (-28.65) |
| GDP | -0.0000*** | -0.0000*** | -0.0000*** | -0.0000*** | -0.0000*** | -0.0000*** | -0.0000*** | -0.0000*** | -0.0000*** | -0.0000*** | -0.0000*** | -0.0000*** |
| | (-40.47) | (-60.89) | (-81.95) | (-58.16) | (-58.18) | (-83.13) | (-57.61) | (-58.80) | (-83.22) | (-73.95) | (-25.86) | (-82.64) |
| _cons | -2.4882*** | 0.5995*** | 2.0437*** | -2.3788*** | -3.7582*** | 2.3541*** | 3.2623*** | 2.4469*** | 2.3160*** | 2.1693*** | 2.6563*** | 2.7989*** |
| | (-52.22) | (6.28) | (31.37) | (-47.43) | (-79.34) | (28.99) | (33.77) | (27.87) | (34.79) | (28.90) | (20.49) | (43.07) |
| Year | YES | YES | YES | YES | YES | YES | YES | YES | YES | YES | YES | YES |
| Ind | YES | YES | YES | YES | YES | YES | YES | YES | YES | YES | YES | YES |
| N | 16,502 | 25,062 | 41,564 | 18,893 | 22,671 | 41,564 | 19,004 | 22,560 | 41,564 | 34,534 | 7,030 | 41,564 |

Note:*,**and***represent significance levels at 0.1,0.05 and 0.01, respectively.

coefficient of the interaction of company misconduct and company location dummy variable (*Violate*Violate*ProvNum*) is 0.4691 and significant at the 1% level. These results indicate that the survival risk of eastern company is higher than that of central-western company after misconduct.

## Discussion

The above theoretical analyses and empirical rests reveal that company misconduct is a threatening factor to company survival, and company misconduct has negative effects on the survival through the mechanism of investor confidence and financing constraints. Existing studies have pay considerable attention to various influencing factors of company survival like firm size, age, ownership structure,innovation, export, etc. The analysis results presented in part four provides a supplement to the influencing factors of company survival. Furthermore, some interesting phenomenons are found through the heterogeneity analysis.

Companies with different ownership structures have different governance structures, organizational systems, and market positions. As a result, their motivations for misconduct are different. Scholars study the relationship between ownership and misconduct in Chinese listed company from the perspective of controlling shareholders, and find that compared to state-owned company, non-state-owned company exhibits more severe misconduct [95, 96]; state-owned company has a lower probability of being punished for misconduct than non-state-owned company [97], and non-state-owned company face stricter loan restrictions after misconduct than state-owned enterprises [98]. As the result shows in Table 8, the state-owned company has a lower survival risk than non-state-owned company after misconduct, it could be explained by the fact that, compared to non-state-owned company, state-owned company can receive more policy support and have closer relationships with financial institutions. Even in the case of misconduct, the government and financial institutions continue to provide policy support and try their best to help state-owned company overcome difficulties, thus facing lower survival risk.

The size of a company to some extent reflects its current development stage, industry level, and market position, leading to differences in investment and financing opportunities. As is shown in Table 8, the survival risk of large-size company is lower than that of small-size company after misconduct. It's probably because large company have a higher market position, greater brand influence, more financing channels, and stronger risk resistance capabilities. Therefore, after misconduct, the large enterprise can fully utilize its own advantages such as brand, resources, and funds to better prevent and resolve the production and operation risks caused by misconduct.

Listed companies have governance heterogeneity, and it causes differences in their governance structures and levels, which leads to various motives and methods of misconduct, and the results are heterogeneous. The equity structure in corporate governance reflects the distribution of control rights within a company, and different equity structure reflect different types of principal-agency relationships, thereby resulting in varying governance performance. Researches show that there is a widespread existence of pyramid holding companies around the world [99–101]. The ultimate controllers exert control over the companies through cross-shareholding, pyramid-shareholding, or excessive control rights, and have a significant impact on the organizational structure, strategic and operational decision-making of the companies. Some scholars believe that equity concentration in company can make it easier for shareholders to monitor, reduce managerial opportunism, mitigate agency conflicts, reduce agency costs, improve corporate performance, enhance corporate value, increase R&D investment, and promote digital transformation of companies [102–104]. However, other scholars hold the

opposite views, arguing that equity concentration is detrimental to corporate governance. Large shareholders can use their information advantages to increase the degree of information asymmetry within and outside the company, which damages the interests of other shareholders. The higher the concentration of equity in listed companies, the more severe the second type of agency problem, and equity balance is beneficial in restricting the interests encroachment behavior of large shareholders [105–107]. The empirical results in Table 8 suggest that high equity concentration company has a lower survival risk compared to low equity concentration company after misconduct. It's probably because decision-making authority in high equity concentration company is more centralized, which is more advantageous for decision-makers to utilize ownership to enhance organizational and managerial capabilities, thus making more efficient decisions and reducing the survival risk after misconduct.

There are significant differences between the eastern and central-western regions of China in terms of marketization, economic development, and legal system. Generally speaking, the eastern region has a higher level of marketization, a more complete legal system, and a higher degree of financial development than central-western region. Even the high level of market competition and legal system in the eastern region can play a governance role to some extent, reducing agency costs for companies [108]. The empirical results in Table 8 show that the survival risk of eastern company is higher than that of central-western company after misconduct. However, it can be seen from Table 3 and column (12) in Table 8 that the coefficients of location dummy variable are negative and significant, which means that the survival rate of the company in the eastern region is higher than that of the company in the central-western region. Thus, it can be concluded that regional factor is a protective factor to company survival, but while company misconduct, the eastern region becomes a threatening factor to company survival. It's probably because the market competition and legal system in the eastern region do not effectively govern misconduct company, a higher legal system level may lead to stricter punishment and supervision of local misconduct company, which makes the survival environment for misconduct company in the eastern region more severe and increasing its operational risks.

## Summary and recommendations

In 2019, when the Covid-19 pandemic began, many enterprises in China have suspended business activities and fallen into crisis. Existence studies focus on the various impact factors on company survival risk like company age, size, innovation, ESG performance etc., but little attention is paid to company misconduct. As such, this paper takes listed companies in China's A-share market from 2000 to 2022 as the research sample, and uses KM estimation and Clog-log discrete time survival model to explore how company misconduct affect company survival risk.The main findings of this paper are as follows. First, company misconduct reduce the survival time of companies, and the survival rate of company with misconduct is significantly lower than that of company without misconduct. Second, company misconduct will increase the survival risk of company through two transmission mechanisms: reducing investor confidence and increasing financing constraints. Thirds, the negative effect of company misconduct on survival rate is more pronounced in non-state-owned companies, small-scale companies, companies with low equity concentration, and companies in the eastern region.

Based on the above research conclusions, the recommendations could be proposed as follows. First, Chinese government should pay attention to prevent illegal and irregular behaviors of listed companies. Misconduct of listed company not only affects the stable development of the capital market, but also reduces the survival rate of company, it is risk factors for the survival and development of the company. Second, the financing ability for company is very

important after misconduct. After being penalized for misconduct, the company face restricted financing channels and increased financing costs, which lead to survival risks for company. Therefore, company with misconduct should actively take measures to restore and enhance its reputation. It can be achieved by strengthening internal management, participating in social welfare and charity activities, which improves company's social image and social responsibility. Third, the government and financial institutions should pay attention to company differentiation when regulating and assisting miscount company. Company of non-state-owned, small-size, low equity concentration, and located in eastern region face more serious survival risk after miscount, thus need more support form the government and financial institutions.

This study has the following limitations:(1) The impact mechanisms of misconduct on company survival risk may vary, this study only examines financing constraints and investor confidence; (2) Different type of companies face different survival risk. This paper only focuses on listed companies' survival risk, other type of companies like small and medium companies' survival risk remains being studied. The above limitations do not affect the accuracy of the conclusions and inspire further study.

## Supporting information

**S1 Data.**
(ZIP)

## Author Contributions

**Conceptualization:** Ling Lu.

**Data curation:** Ling Lu.

**Formal analysis:** Ling Lu.

**Investigation:** Ling Lu.

**Methodology:** Ling Lu.

**Project administration:** Ling Lu.

**Writing – original draft:** Ling Lu.

**Writing – review & editing:** Ling Lu.

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
