## [Decision Letter · Decision Letter 0]

7 May 2024

PONE-D-24-10269Does company misconduct affect company survival risk?— evidence from ChinaPLOS ONE

Dear Dr. Lu,

Thank you for submitting your manuscript to PLOS ONE. After careful consideration, we feel that it has merit but does not fully meet PLOS ONE’s publication criteria as it currently stands. Therefore, we invite you to submit a revised version of the manuscript that addresses the points raised during the review process. One reviewer is very critical about your manuscript, the other less so. However, take all of the reviewers' comments into very careful consideration when revising your manuscript.

We look forward to receiving your revised manuscript.

Kind regards,

Robin Haunschild

Academic Editor

PLOS ONE

“National Natural Science Foundation of China (72263021); China Postdoctoral Science Foundation (2020M682104); University Humanities and Social Science Research Projects of Jiangxi Province(JJ22210).”

Reviewers' comments:

Reviewer's Responses to Questions

**Comments to the Author**

1. Is the manuscript technically sound, and do the data support the conclusions?

Reviewer #1: No

Reviewer #2: Yes

2. Has the statistical analysis been performed appropriately and rigorously? 

Reviewer #1: No

Reviewer #2: Yes

3. Have the authors made all data underlying the findings in their manuscript fully available?

Reviewer #1: No

Reviewer #2: No

4. Is the manuscript presented in an intelligible fashion and written in standard English?

Reviewer #1: No

Reviewer #2: Yes

5. Review Comments to the Author

Reviewer #1: Dear authors and editor,

Many thanks for inviting me to review the manuscript titled “Does company misconduct affect company survival risk?— evidence from China” (#PONE-D-24-10269) submitting to Plos One (ISSN / eISSN: 1932-6203).

After reading it, I should point out some issues

1.The authors should pay more attention to the tense through the whole manuscript.

2.In the Keywords, they are just so-so and the authors keep the order from the capital letter "A-Z" and the number is 4 or 5 preferred.

3.In the Introduction, the materials of the paper should be just update to 2024, and the authors also can look through the recent and important paper on this topic again. It lacks important literature. In the Literature section, we should link to the recent and most literature, specially 2021-2023. Besides, we also pay more attention to the value and contribution of the paper. Besides, I can not see the gap of the literature.

4.In the section of “Literature review and theoretical hypothesis”, also the recent and important references should be updated.

5.The fig of research model is missing and meanwhile its value is limited.

6.In the section of Participants and design, the authors should add more details into the section.

7.The authors should add the sources into “Table 1. Variable Definitions and Calculation Methods”.

8. Before the conclusion section, the authors should adjust discussion section, comparing the findings of the study with previous studies. It is very important and valuable. However, the present version is missing .The authors should add it.

9. The section of conclusion is not good and the authors should add more into the section. After the conclusion section, the authors should add limitation section.

10.The authors should check all the language over the whole manuscript.

11. In terms of references, the current number quality of references is relatively limited. Additionally, it is important to ensure that the format meets the requirements of Plos One .

Therefore, based on the value and contribution of the present version, I only choose “Reject”, and also welcome the revised submission in the future. Good luck !

All the best

April 24, 2024

Reviewer #2: The study investigates the impact of company misconduct on company survival risk through survival analysis. The study has conducted a battery of robustness check and obtained consistent results in all. The study also pertains to potential underlying mechanism. However, the motivation of the study is not sufficiently elaborated. Why investigate company survival risk? There are other topic like ESG to exploit. It is advised that the author substantiate the motivation. Besides, although the paper is written in understandable English, it is advised that the paper had better be proofread before publication. A minor issue is in heterogeneity tests, coefficients cannot be directly compared. Phisher's permutation test is recommended.

6. PLOS authors have the option to publish the peer review history of their article (what does this mean?). If published, this will include your full peer review and any attached files.

Reviewer #1: No

Reviewer #2: No

---

## [Author Response · Author response to Decision Letter 0]

9 Jun 2024

Jun 9, 2024

Dear Editor:

Thank you very much for giving me an opportunity to revise my manuscript, I appreciate reviewers very much for their constructive comments and suggestions on my manuscript entitled “How does company misconduct affect company survival risk?— evidence from China”(ID:PONE-D-24-10269). Those comments are all valuable and very helpful for revising and improving my paper, as well as the important guiding to my researches. I have studied comments carefully and made corrections, and I hope these corrections meet criteria. Revised portions are marked in blue in the paper. According to your and reviewer’s comments, I have made the following major corrections:

(1)I have reorganized the introduction, and made it meet reviewers suggestions;

(2)I have reorganized the literature and made some supplements;

(3)I have added the discussion part in the paper; 

(4)I have added limitation section in the “Summary and recommendations”part;

(5)I have adjusted the key words in the abstract.

Additional, I have revised the language to make it more clear and concise. The specific responds to the review’s comments are in the following pages. The comments are in italic and responds are in orthography. Please review the revised version and let us know if any further modifications are required. I am grateful for your thorough review and look forward to you feedback.

Thank you once again for your time and expertise.

Yours sincerely,

Ling Lu

Corresponding author: Ling Lu

E-mail: lulu_995179@163.com

Reviewer #1:

1.Review’s comment: the authors should pay more attention to the tense through the whole manuscript.

Response: I thank the referee for pointing out my incorrect writing in the tense. I mixed present and past tense in the literature review, thus I have revised the tense into present.

2.Review’s comment: they are just so-so and the authors keep the order from the capital letter "A-Z" and the number is 4 or 5 preferred.

Response: Keywords should reflect the main content of the paper. According to this principal, I have re-screened the keywords and arranged in alphabetical order on page2.

3.Review’s comment: In the Introduction, the materials of the paper should be just update to 2024, and the authors also can look through the recent and important paper on this topic again. It lacks important literature. In the Literature section, we should link to the recent and most literature, specially 2021-2023. Besides, we also pay more attention to the value and contribution of the paper. Besides, I can not see the gap of the literature.

Response: I appreciate these good suggestions. First, as the previous material has little relevance to the main content of the paper, I have deleted this part and re-written the background of the paper in the first paragraph of “Introduction” part on page 2. Second, I have added a brief literature review in the second paragraph of “Introduction” part on page 2. Based on the brief review, I have reorganized the literature and added the topics on the connotation and measurement of company survival, and key determinants of company survival from page 3 to 5. Third, according to the literature review and innovations of the paper, I have emphasized the contributions in the fourth paragraph of “Introduction” part on page 2. Fourth, I have summarized the gap in the first paragraph of “Gaps and research hypotheses” part from page 5-6.

4.Review’s comment: In the section of “Literature review and theoretical hypothesis”, also the recent and important references should be updated.

Response: I have updated references and added the relevant topic in the “literature review” part from page 3-5.

5.Review’s comment: The fig of research model is missing and meanwhile its value is limited.

Response: Thank you for pointing out this problem in manuscript. I have rechecked the manuscript and made supplement on missing model and value. 

6.Review’s comment: In the section of Participants and design, the authors should add more details into the section.

Response: Thank you for your rigorous comment. This section was aimed to introduce the data and method instead of research design, thus I have changed this inaccurate description into “Data and method”.

7.Review’s comment: The authors should add the sources into “Table 1. Variable Definitions and Calculation Methods”.

Response: I have added data sources into Table 1, and kept all variable symbols in italic.

8.Review’s comment: Before the conclusion section, the authors should adjust discussion section, comparing the findings of the study with previous studies. Response: Considering the Reviewer's suggestion, I have added discussion section from page 24 to 25, mainly on heterogeneity analysis.

9.Review’s comment: The section of conclusion is not good and the authors should add more into the section. After the conclusion section, the authors should add limitation section.

Response: According to the Reviewer's comments, I have re-written this part, and reorganized summary and recommendations sections, and added limitation section on page 26.

10.Review’s comment: The authors should check all the language over the whole manuscript.

Response: I am very sorry about for my incorrect writing in language, and I have revised the language to make it more clear and concise.

11.Review’s comment:In terms of references, the current number quality of references is relatively limited. Additionally, it is important to ensure that the format meets the requirements of Plos One .

Response: I have reorganized references and revised the format to ensure that it meets the requirements of Plos One. 

Reviewer #2: The motivation of the study is not sufficiently elaborated. Why investigate company survival risk? There are other topic like ESG to exploit. It is advised that the author substantiate the motivation. Besides, although the paper is written in understandable English, it is advised that the paper had better be proofread before publication. A minor issue is in heterogeneity tests, coefficients cannot be directly compared. Phisher's permutation test is recommended.

Response: Thank you for your comments. Considering the Reviewer's comments, I have reorganized the literature, and emphasized the motivation in the fourth paragraph of “Introduction” part on page 2 and supplied the gap in the first paragraph of “Gaps and research hypotheses” part from page 5-6. About English writing, I have revised the language to make it more clear and concise. Additional, there are many ways for heterogeneity tests, and Phisher's permutation test is really a good way. I have used the grouped regression and full-sample interaction regression to examine the heterogeneity, and they achieved the same results.

---

## [Decision Letter · Decision Letter 1]

25 Jun 2024

How does company misconduct affect company survival risk?— evidence from China

PONE-D-24-10269R1

Dear Dr. Lu,

We’re pleased to inform you that your manuscript has been judged scientifically suitable for publication and will be formally accepted for publication once it meets all outstanding technical requirements.

Kind regards,

Robin Haunschild

Academic Editor

PLOS ONE

Additional Editor Comments (optional):

Reviewers' comments:

Reviewer's Responses to Questions

**Comments to the Author**

1. If the authors have adequately addressed your comments raised in a previous round of review and you feel that this manuscript is now acceptable for publication, you may indicate that here to bypass the “Comments to the Author” section, enter your conflict of interest statement in the “Confidential to Editor” section, and submit your "Accept" recommendation.

Reviewer #2: All comments have been addressed

Reviewer #3: All comments have been addressed

2. Is the manuscript technically sound, and do the data support the conclusions?

Reviewer #2: Yes

Reviewer #3: Yes

3. Has the statistical analysis been performed appropriately and rigorously? 

Reviewer #2: Yes

Reviewer #3: Yes

4. Have the authors made all data underlying the findings in their manuscript fully available?

Reviewer #2: Yes

Reviewer #3: Yes

5. Is the manuscript presented in an intelligible fashion and written in standard English?

Reviewer #2: Yes

Reviewer #3: Yes

6. Review Comments to the Author

Reviewer #2: (No Response)

Reviewer #3: Since all the comments are revised by the authors, this study can be accepted for publication. Good luck!

7. PLOS authors have the option to publish the peer review history of their article (what does this mean?). If published, this will include your full peer review and any attached files.

Reviewer #2: No

Reviewer #3: No

---

## [Editor Report · Acceptance letter]

5 Jul 2024

PONE-D-24-10269R1 

PLOS ONE

Dear Dr. Lu, 

I'm pleased to inform you that your manuscript has been deemed suitable for publication in PLOS ONE. Congratulations! Your manuscript is now being handed over to our production team.

Kind regards, 

on behalf of

Dr. Robin Haunschild 

Academic Editor

PLOS ONE